# Rapid COVID-19 Molecular Diagnostic System Using Virus Enrichment Platform

**DOI:** 10.3390/bios11100373

**Published:** 2021-10-06

**Authors:** Yoon Ok Jang, Hyo Joo Lee, Bonhan Koo, Hye-Hee Cha, Ji-Soo Kwon, Ji Yeun Kim, Myoung Gyu Kim, Hyun Soo Kim, Sung-Han Kim, Yong Shin

**Affiliations:** 1Department of Biotechnology, College of Life Science and Biotechnology, Yonsei University, Seoul 03722, Korea; jangyo17@daum.net (Y.O.J.); hyoj0125@gmail.com (H.J.L.); qhsgksdlek@naver.com (B.K.); wws94@naver.com (M.G.K.); 2Department of Infectious Diseases, Asan Medical Center, University of Ulsan College of Medicine, Songpa-gu, Seoul 05505, Korea; heyhe0102@naver.com (H.-H.C.); kwonjs92@kaist.ac.kr (J.-S.K.); aeki22@snu.ac.kr (J.Y.K.); 3Department of Convergence Medicine, Asan Medical Institute of Convergence Science and Technology (AMIST), University of Ulsan College of Medicine, Songpa-gu, Seoul 05505, Korea; 4INFUSIONTECH, 38, Heungan-daero 427 beon-gil, Dongan-gu, Anyang-si 14059, Korea; hskim@infusiontech.co.kr

**Keywords:** COVID-19, molecular diagnostics, sample preparation, virus enrichment, rapid diagnostics

## Abstract

The coronavirus disease 2019 (COVID-19) pandemic, caused by the severe acute respiratory syndrome coronavirus (SARS-CoV)-2, is rapidly spreading and severely straining the capacities of public health communities and systems around the world. Therefore, accurate, rapid, and robust diagnostic tests for COVID-19 are crucial to prevent further spread of the infection, alleviate the burden on healthcare and diagnostic facilities, and ensure timely therapeutic intervention. To date, several detection methods based on nucleic acid amplification have been developed for the rapid and accurate detection of SARS-CoV-2. Despite the myriad of advancements in the detection methods for SARS-CoV-2, rapid sample preparation methods for RNA extraction from viruses have rarely been explored. Here, we report a rapid COVID-19 molecular diagnostic system that combines a self-powered sample preparation assay and loop-mediated isothermal amplification (LAMP) based naked-eye detection method for the rapid and sensitive detection of SARS-CoV-2. The self-powered sample preparation assay with a hydrophilic polyvinylidene fluoride filter and dimethyl pimelimidate can be operated by hand, without the use of any sophisticated instrumentation, similar to the reverse transcription (RT)-LAMP-based lateral flow assay for the naked-eye detection of SARS-CoV-2. The COVID-19 molecular diagnostic system enriches the virus population, extracts and amplifies the target RNA, and detects SARS-CoV-2 within 60 min. We validated the accuracy of the system by using 23 clinical nasopharyngeal specimens. We envision that this proposed system will enable simple, facile, efficient, and inexpensive diagnosis of COVID-19 at home and the clinic as a pre-screening platform to reduce the burden on the medical staff in this pandemic era.

## 1. Introduction

The pandemic of coronavirus disease 2019 (COVID-19) has emerged as a life-threatening respiratory disease caused by the severe acute respiratory syndrome coronavirus 2 (SARS-CoV-2) [1,2]. COVID-19 is rapidly spreading around the world due to sustained human-to-human transmission. Consequently, COVID-19 has substantially overwhelmed global healthcare and health resources due to the rapidly increasing number of suspected patients [3,4,5]. The presentation of this disease among individuals is highly heterogeneous in severity, ranging from asymptomatic to highly morbid cases [6,7]. Both symptomatic and asymptomatic patients are highly contagious and transmit the disease to many individuals, thereby acting as “super infectors”, before being identified as active carriers of SARS-CoV-2 [8,9]. In particular, there is currently no optimal therapeutics available to treat COVID-19. Thus, the management of this pandemic currently relies on its diagnosis. As a consequence, healthcare workers and researchers in healthcare and diagnostic facilities have been facing the overwhelming burden of testing large populations for COVID-19 diagnosis. Moreover, there has been an increase in the number of asymptomatic infected patients. Thus, a rapid, accurate, robust, and self-testing procedure for COVID-19 diagnosis is highly desirable [10] and urgently required to alleviate the burden on hospital and laboratory facilities, ensure a timely therapeutic intervention, and prevent further spread of epidemic and pandemic infectious diseases [11,12,13]. To overcome current issues related to the diagnosis of COVID-19, widespread rapid diagnostic testing is desired that allows efficient enrichment of the virus titer, extraction, and detection, all of which are crucial for the diagnosis and disease surveillance of SARS-CoV-2. Awareness about the current situation of COVID-19 in the world can be increased by developing a novel technique that introduces a pre-screening step at home or in small clinics, which would decrease the workload on healthcare workers and researchers. Hence, the ultimate purpose is to take early diagnostics directly to individuals around the world in point-of-care (POC) testing, requiring a self-sample preparation without any sophisticated instrumentation. 

At present, nucleic acid (NA)-based testing is used as the main diagnostic strategy of COVID-19 [14,15,16,17]. The quantitative polymerase chain reaction (qPCR) is the most common and widely used technology for the quantitative and qualitative assessment of NAs [18,19,20]. Although the qPCR assay is sensitive and specific, it is relatively time-consuming and requires specific equipment and devices, as well as expensive reagents. These drawbacks render qPCR analysis incapable to meet the recent huge demand for the diagnosis of COVID-19 cases. Therefore, alternative isothermal amplification methods, including recombinase-based polymerase amplification (RPA), helicase-dependent amplification (HDA), rolling-circle amplification, and loop-mediated isothermal amplification (LAMP) have emerged as important diagnostic methods used in basic scientific research and clinical applications [21,22,23,24,25,26,27,28,29]. Among these technologies for use of COVID-19 detection, LAMP has been applied to the POC of detecting pathogens, including bacteria [30,31], viruses [32,33], malaria [34,35,36], and fungi [37]. Moreover, LAMP assays are simple, rapid, and do not require bulky instruments, expensive detergents, or highly trained professionals [38,39]. Furthermore, LAMP technologies for the detection of RNA viruses have been rapidly developing [21,22,23,40]. Current modern optical tools offer a facile, fast, and inexpensive approach for the sensitive and specific detection of pathogens based on visible-color development, fluorescence, or chemiluminescence. Specifically, the lateral flow assay (LFA) has been widely used in viral detection [41]. The colorimetric reverse transcription (RT)-LAMP assay expands on the basic LAMP technology and involves a one-pot reaction that is based on visual detection of NA amplification [42]. This detection is achieved by the co-presence of reverse transcriptase, DNA polymerase, and a pH indicator dye in the master mix, thereby obviating the need for detection equipment. The LFA and colorimetric assays are portable, and results are observable with the naked-eye [42,43,44]. Therefore, these methods can be used for diagnostic purposes in regions that lack instrumentation, at POC centers, or in the field.

In contrast to the detection methods, sample preparation methods have rarely been explored for COVID-19 case detection. Traditional pathogen sample preparation methods include enrichment and extraction, which are widely used in clinical diagnosis, and which are based on density gradient centrifugation and culture [45,46]. However, these methods pose various constraints, such as contamination and pathogen spread and loss, the requirement of sophisticated instrumentation and expensive reagents, and a labor-intensive process [47,48]. Pathogen concentrations are extremely low at the early stages of the disease, and thus highly specific and sensitive detection methods are needed for early diagnosis and effective surveillance. Above all, pathogen enrichment methods based on sample preparation are of significant importance because they highly correlate with the sensitivity of pathogen detection in clinical applications. Furthermore, the diagnostic methods for COVID-19 remain inefficient in sensitivity and require additional steps, including electrophoresis and fluorescent dye labeling for detection.

In this study, we report a rapid COVID-19 molecular diagnostic system combining a simple virus enrichment and extraction method that involves dimethyl pimelimidate (DMP), a hydrophilic polyvinylidene fluoride (PVDF) filter, and a LAMP-LFA method. This proposed system could provide a sensitive, facile, efficient, and inexpensive method for the diagnosis of COVID-19. The combination of the RT-LAMP assay and DMP-PVDF filter for virus enrichment, amplification, and detection of viral RNA results in increased sensitivity and specificity while reducing the contamination risk and complicated processes. The proposed system detects viral RNA at a dilution of 2.33 × 10^2^ copies per reaction. We validated the clinical utility of the rapid COVID-19 molecular diagnostic system by testing 23 clinical nasopharyngeal specimens. Using this system, SARS-CoV-2 from human specimens was simultaneously enriched, amplified, and detected at high sensitivity within 60 min. Therefore, the proposed COVID-19 diagnostic system is suitable for rapid, facile, and simple diagnosis at inadequately equipped or under-staffed healthcare centers and in situations where results are urgently required for the optimal diagnosis of emerging infectious diseases.

## 2. Materials and Methods

### 2.1. Bacterial Sample

To investigate the capacity of DMP and PVDF filter assays for bacterial cells, we used the extracted DNA from bacteria cells of *Escherichia coli (E. coli)* and isolated RNA from *Brucella ovis*. *E. coli* (ATCC 25922) was inoculated in nutrient broth medium and incubated overnight at 37 °C under shaking conditions. *Brucella ovis* (ATCC 25840) was grown in Brucella agar containing 5% defibrinated sheep blood and incubated at 37 °C in a 5% CO_2_ atmosphere for 48 h.

### 2.2. VeroE6 Cells Infected with SARS-CoV-2

To identify the capacity of DMP and PVDF filter assays for the quantitation of SARS-CoV-2 viral load, viral RNA was extracted from the culture fluid of SARS-CoV-2-infected VeroE6 cells (Zeptometrix, Cat. No. 0810590CFHI). SARS-CoV-2 viral loads were quantified using an RT-qPCR assay targeting the S and N genes. The RT-qPCR conditions for RNA were as follows: RT at 50 °C for 10 min; an initial denaturation step at 95 °C for 30 s; 40 cycles at 95 °C for 5 s, 60 °C for 30 s; and cooling steps at 40 °C for 30 s. The RT-qPCR was performed on a CFX96 Touch Real-Time PCR Detection System (Bio-Rad, Hercules, CA, USA) using 5 μL RNA and the LightCycler^®^ Multiplex RNA Virus Master (Roche, Germany). The sequences of the primers and probes are presented in Appendix A.

### 2.3. Generating Transcribed T7 RNA and Clinical Specimen Collection

To evaluate the limit of detection the RT-LAMP assays, transcribed T7 RNA was generated with SARS-CoV-2. To prepare the RNA standard, the spike (S) and nucleocapsid (N) genes (GenBank MN908947.3) of SARS-CoV-2 were amplified from a positive clinical sample with primers containing the T7 promoter sequence. Detailed primer sequences for the RNA synthesis are presented in Appendix A. The T7-flagged PCR product was synthesized for transcription using a MEGAscript kit (Thermo Fisher Scientific Solutions LLC, Seoul, Korea). Then, RNA was purified with a MEGAclear kit (Thermo Fisher Scientific Solutions LLC, Seoul, Korea) following the manufacturer’s instructions. The RNA concentration and quality were assessed using a NanoDrop UV-vis spectrophotometer (Thermo Fisher Scientific Solutions LLC, Seoul, Korea). For quantification of the RNA transcript, the RNA copy number was calculated using the EndMemo DNA/RNA copy number calculator (http://endmemo.com/bio/dnacopynum.php; 13 May 2021). The nasopharyngeal specimens (1 mL) were collected from patients infected with human SARS-CoV-2, human coronavirus (HCoV)-OC43/229E, and healthy controls who agreed to sampling and were admitted to the Asan Medical Center (AMC), Republic of Korea. This study was reviewed and approved by the ethics committee of the institutional review board of the Asan Medical Center, and all participants gave written informed consent. All experimental procedures were performed following the guidelines approved by the Institutional Review Board of the AMC (IRB No. 2020-0297). The nasopharyngeal swabs were stored in a universal transport medium (Noble Biosciences, Seoul, Korea). Clinical specimens were inactivated by heating at 60 °C for 30 min and then stored at −80 °C until analysis.

### 2.4. Enrichment of Pathogens and Characterization of DMP and PVDF Filter

The procedure for pathogen enrichment was as follows: (1) 300 μL of DMP (100 mg/mL) was pipetted into 1 mL bacteria sample solution. The mixture was transferred into the ABLUO Syringe PVDF filter (0.45 μm, 13 mm, GVS). (2) The mixture was incubated for 10 min at room temperature to enable pathogen capture. (3) The PVDF filter was washed with 1 mL PBS to remove debris and uncaptured pathogens. (4) The enriched pathogen solution was collected from the PVDF filter using an elution buffer consisting of 10 mM sodium bicarbonate (NaHCO_3_, pH 10.6). The interaction of NAs and DMP on a PVDF membrane was characterized using Fourier-transform infrared (FTIR) spectroscopy (JASCO 6300, JASCO, Easton, MD, USA).

### 2.5. Nucleic Acid Extraction (NA)

The process for NA extraction was as follows: (1–1) NA extraction from the enriched pathogen solution with *E. coli* and *Brucella* was added to 150 μL lysis buffer (100 mM Tris-HCl (pH 8.0), 10 mM EDTA, 1% SDS, 10% Triton X-100), 150 μL DMP (100 mg/mL), and 20 μL Proteinase K (Qiagen, Germany). Selectively, for RNA extraction, the lysis buffer was added to 30 μL of DNase (Qiagen). (1–2) The mixture was transferred into the PVDF filter and incubated for 10 min at 56 °C or room temperature for DNA and RNA extraction, respectively. (1–3) Afterward, the PVDF filter was washed with 1 mL PBS. (1–4) Subsequently, the NA was extracted from the PVDF filter using the elution buffer. (2) The RNA of the enriched pathogen solution from 100 μL clinical specimen and 100 μL SARS-CoV-2 culture fluid was extracted using the QIAamp Viral RNA Mini Kit (Qiagen) according to the manufacturer’s instructions.

### 2.6. Reverse Transcription-Polymerase Chain Reaction (RT-PCR)

RT-PCR and RT-qPCR were performed to determine the efficiency of the pathogen enrichment and NA extraction with different filter membranes, membrane types, and membrane pore sizes. The primer sequences are listed in Appendix A. The RT-PCR conditions for RNA were as follows: RT at 50 °C for 30 min; an initial denaturation step at 95 °C for 15 min; 45 cycles at 95 °C for 30 s, 58 °C for 30 s, and 72 °C for 45 s; and an extension step at 72 °C for 10 min. The RT-PCR was performed on a T100TM Thermal Cycler (Bio-Rad, Hercules, CA, USA) using 5 μL of RNA and Hyperscrit™ One-Step RT-PCR Master Mix (GeneAll, Seoul, Korea). The PCR conditions for DNA were as follows: an initial denaturation step at 95 °C for 15 min; 45 cycles at 95 °C for 30 s, 58 °C for 30 s, and 72 °C for 30 s; and an extension step at 72 °C for 5 min. The PCR was performed on a T100TM Thermal Cycler (Bio-Rad, CA, USA) using 5 μL of DNA and Taq PCR Core Kit (Qiagen, Hilden, Germany). The RT-qPCR cycling conditions were as follows: RT at 50 °C for 20 min; an initial denaturation step at 95 °C for 15 min; 40 cycles at 95 °C for 10 s, 58 °C for 20 s, and 72 °C for 20 s; and melting steps at 95 °C for 30 s, 65 °C for 30 s, and 95 °C for 30 s. Amplification reactions containing 5 μL of RNA were performed with AriaMx SYBR Green RT-QPCR Master Mix (Agilent Technology, CA, USA) in a CFX96 Touch Real-Time PCR Detection System (Bio-Rad, CA, USA) in accordance with the manufacturer’s instructions. The qPCR cycling conditions were as follows: an initial denaturation step at 95 °C for 15 min; 40 cycles at 95 °C for 10 s, 58 °C for 20 s, and 72 °C for 20 s; and melting steps at 95 °C for 30 s, 65 °C for 30 s, and 95 °C for 30 s. Amplification reactions containing 5 μL of DNA were performed with AriaMx SYBR Green QPCR Master Mix (Agilent Technology, CA, USA) in a CFX96 Touch Real-Time PCR Detection System (Bio-Rad, CA, USA) in accordance with the manufacturer’s instructions.

### 2.7. Isothermal Amplification Assays

The RPA reaction was performed using the 3 μL RNA and a TwistAmp^®^ RT Basic kit (TwistDX, Cambridge, UK); the incubation lasted 25 min at 40 °C on a T100TM Thermal Cycler (Bio-Rad). RPA products were analyzed on a 2% agarose gel and with an LFA using the Milenia HybriDetect 1 kit (TwistDx). To evaluate the RT-thermophilic HDA assays, we used the IsoAmp^®^ II Universal tHDA Kit (New England Biolabs, Beverly, MA, USA). The HDA reaction was performed using a mixture of 5 μL RNA, primers, 25 μL reaction buffer (10X PCR Buffer, MgSO4 (100 mM), NaCl (500 mM), dNTP, IsoAmp enzyme), and ProtoScript ll RT enzyme (PROMEGA); incubation lasted 2 min at 42 °C and 90 min at 65 °C on a T100TM Thermal Cycler (Bio-Rad). The HDA products were analyzed on a 2% agarose gel and with an LFA using the Milenia HybriDetect 1 kit (TwistDx).

### 2.8. RT-LAMP Assays

The RT-LAMP primers targeting the S and N genes of SARS-CoV-2 (GenBank MN908947.3) were designed using the PrimerExplorer V5 software (http://primerexplorer.jp/e/; 3 December 2020). Primer sets included a forward outer primer (F3), backward outer primer (B3), forward inner primer (FIP), backward inner primer (BIP), loop forward primer (LF), and loop backward primer (LB). For RT-LAMP with LFA, the 5′-end of LF primer was labeled with carboxyfluorescein (FAM) and the 5′-end of LB primer was labeled with biotin. The primers for each LAMP PCR assay method (RT-LAMP, RT-LAMP with LFA, and colorimetric RT-LAMP) are presented in Appendix A in detail. To detect the viral N-gene via the RT-LAMP, a WarmStart LAMP 2X Master Mix (NEB) was mixed with primer solution containing all LAMP primers (F3, B3, FIP, BIP, LF, and LB) and 5 µL RNA templates (extracted from virus or synthesized RNA). Primers were used at final concentrations of 1.6 μM for FIP/BIP, 0.2 μM for F3/B3, and 0.4 μM for LF and LB. The RT-LAMP reactions were performed at 65 °C for 30 or 60 min on a T100^TM^ Thermal Cycler (Bio-Rad). LAMP products were analyzed by electrophoresis on a 2% agarose gel.

### 2.9. Naked-Eye Detection Methods

For colorimetric RT-LAMP detection, the WarmStart Colorimetric LAMP 2X Master Mix (NEB) was mixed with primer solution containing all LAMP primers and 5 µL RNA templates. The reactions were performed at 65 °C for 30 or 60 min, and a color change from red to yellow was observed. For the detection of LAMP-amplified product with LFA using the Milenia HybriDetect 1 kit (TwistDx), WarmStart LAMP 2X Master Mix (NEB) was mixed with LAMP primers and 5 µL RNA templates. A 5 µL sample of RT-LAMP amplification was transferred to a new tube, mixed with 100 µL of an LFA dilution solution (TwistDx), and then placed on a sample pad of a lateral flow strip (TwistDx). Lateral flow strips were immersed in the mixed solution, and amplification was observed and evaluated as either positive (bottom line of the strip) or negative (control line, top line of the strip) after 1~2 min. 

## 3. Results

### 3.1. Principles of Procedure for SARS-CoV-2 Enrichment and Detection

The principles underlying the rapid COVID-19 molecular diagnostic system for the enrichment and detection of SARS-CoV-2 are illustrated in Figure 1. This system consists of the RT-LAMP combined with the DMP reagent and PVDF filter. The enhanced pathogen enrichment process is achieved by the use of the DMP-PVDF filter. DMP consists of methylene groups and bifunctional imidoester groups. The positively charged DMP, whose charge results from the amino group, directly captured the negatively charged virus via electrostatic adsorption (Step 1). For pathogen enrichment, the various types of samples were mixed with DMP, and the mixture was then transferred into the PVDF filter. The mixture was incubated for 10 min at room temperature to allow the viruses to bind to DMP via covalent bonding and electrostatic coupling on the PVDF filter surface by hand (Step 1). The PVDF filter was then washed with 1 mL PBS to remove debris and any uncaptured pathogen. The enriched pathogen solution was collected from the PVDF filter using the elution buffer. Subsequently, NA from the enriched pathogen solution was extracted. RNA from the enriched virus solution from clinical specimens was extracted using the spin-column kit. Then, the RNA extracted was amplified using RT-LAMP either with LFA or colorimetric assays that allowed the naked-eye detection of SARS-CoV-2. Therefore, we confirmed that RNA was successfully extracted in both procedures by downstream analysis (Step 2). Our results show that the COVID-19 molecular diagnostics system not only allows for pathogen enrichment but also for the simple and rapid detection of pathogens within 60 min.

### 3.2. Characterization and Application of the DMP-PVDF Filter for Sample Preparation

To test the usability of the sample preparation platform of the DMP-PVDF filter for pathogen enrichment and extraction, we used *E. coli* and *Brucella ovis* as samples for testing pathogen enrichment and extraction steps. We demonstrated the efficiency of the DMP-PVDF filter for pathogen enrichment and extraction using *E. coli* (10^5^ CFU/mL) via RT-qPCR. The positive control was DNA extracted from the absolute concentration *E. coli* (10^5^ CFU) sample using a spin column kit. The RT-qPCR cycle threshold (C_T_) values of the samples enriched using the DMP-PVDF filter were lower than those obtained using the PVDF filter alone (Figure 2A). This result indicates that the DMP increased the efficiency of pathogen enrichment directly as a cross-linker on the PVDF membrane. Subsequently, we assessed the capture efficiency of the filter membrane according to the pore size and diameter. The experiment was repeated using the filter with different membrane pore sizes and diameters (0.2 µm–13 mm, 0.45 µm–13 mm, and 0.45 µm–25 mm). We observed the highest amplification efficiency with the filter with pore diameter 0.45 µm–13 mm (Figure 2B). Furthermore, we evaluated the capture efficiency of various filter membrane materials. The same experiment was conducted using PVDF, Cellulose Acetate (CA), Polyethersulfone (PES), Nylon (NY), Regenerated Cellulose (RC), Polyethylene (PE), Nitrocellulose (NC), Polytetrafluoroethylene-Hydrophilic (PTFE-HP), or PTEE. We observed that the PVDF filter provided the highest amplification efficiency among the filter membrane materials tested (Figure 2C). Moreover, the amplification efficiency of the PVDF filter was similar to that obtained using the spin column kit (Figure 2C). The PTFE-HP filter showed higher amplification efficiency than the PTFE filter. This indicates that PTFE-HP composed of a hydrophilic membrane efficiently captures bacteria and NAs. Thus, these results indicate that the PVDF filter composed of a hydrophilic membrane efficiently capture bacteria and NAs. We further assessed the efficiency of the elution buffer on the DMP-PVDF filter platform. The method was tested using elution buffers of different compositions and pH, such as D.W., AVE buffer (Qiagen kit), 10 mM NaHCO_3_ (pH 10.6), and 10 mM Tris-HCl (pH 8.0). We observed the highest amplification efficiency with an elution buffer containing 10 mM NaHCO_3_ (pH 10.6) on the DMP-PVDF filter platform (Appendix A). After optimizing the DMP-PVDF filter for pathogen enrichment and extraction, we evaluated its capacity to detect *E. coli* for DNA and the *Brucella ovis* for RNA ranging from 1 × 10^2^ to 1 × 10^5^ CFU/mL. The RT-qPCR CT value of the DNA and RNA extracted using the DMP-PVDF filter (light green) was earlier than those obtained using the spin column kit (gray), which extracted DNA and RNA without pathogen enrichment (Appendix A). Moreover, the amplification efficiency of the DMP-PVDF filter platform was similar to the absolute value of the pathogen (Qiagen kit, white) (Appendix A). These results suggest that the capacity of the DMP-PVDF platform with pathogen enrichment is more efficient than those of conventional commercial kits that do not involve pathogen enrichment. Next, we investigated the capacity of the DMP-PVDF platform with pathogen enrichment by using SARS-CoV-2 culture fluid ranging from 1 × 10^1^ to 1 × 10^5^ PFU/100 µL. SARS-CoV-2 viral loads were quantified using RT-qPCR assay for targeting the N and S genes (Figure 2D). The detection limit of the platform is 10 PFU/100 µL for both N and S genes. These results suggest that the DMP-PVDF platform can also be used with SARS-CoV-2 enrichment.

Fourier-transform infrared spectroscopy (FTIR) was used to confirm the interaction of NAs and DMP on a PVDF membrane. The FTIR spectra of the plain PVDF membrane (black line), and the NAs (red line), DMP (blue line), and combination of the NAs and DMP (pink line) on the PVDF membrane are shown in Figure 3. The NAs and DMP combination on the PVDF membrane exhibit a distinct absorption peak at 3392 cm^−1^, which is attributed to the O-H/N-H stretching vibrations [49]. The NAs and DMP combination on the PVDF membrane exhibit a higher intensity of wavenumbers for the characteristic absorptions (CH_2_ asymmetrical stretching variation at 3021 cm^−1^ and 1170 cm^−1^, and CH_2_ symmetrical stretching variation at 2979 cm^−1^) than the PVDF membrane, NAs, or DMP alone [50,51,52]. Moreover, for the NAs and DMP combination on the PVDF, we found that the characteristic absorptions, including CH_2_ wagging, CF_2_ stretching, C-C-C asymmetrical stretching, and C-F stretching variations, occurred at 1402, 1070, 878, and 834 cm^−1^, respectively [49,51,52,53,54]. Furthermore, the characteristic absorptions for the γ phase occurred at 431, 482, and 1233 cm^−1^, and the characteristic absorption for the β and γ phases occurred at 510 cm^−1^ for the NAs and DMP combination on the PVDF (Figure 3) [55]. These results confirm that the FTIR spectrum of DMP on the PVDF membrane is larger than that of the membrane itself due to the relative absorption of DMP. Moreover, the FTIR spectra of the NAs and DMP blends on the PVDF membrane were larger than that of the membrane itself due to the relative absorption of the DMP blends and NAs.

### 3.3. Isothermal Amplification Assays for SARS-CoV-2 Detection

Then, to determine the optimal detection method with the sample preparation method for COVID-19 testing, we demonstrated the sensitivity of various isothermal amplification methods such as RT-RPA, RT-HDA, and RT-LAMP. A ten-fold serial dilution of T7 RNA from the SARS-CoV-2 N gene was used as the template. The RT-RPA, RT-HDA and RT-LAMP were analyzed via either with the paper-based LFA or the colorimetric assay for naked-eye detections and on a 2% agarose gel. The RT-RPA assay was performed using T7 RNA (8.12 × 10^7^ to 8.12 × 10^1^ per reaction). The RT-RPA assay detected viral RNA at a dilution of 8.12 × 10^3^ per reaction with a reaction time of 25 min and a reaction temperature of 40 °C (Figure 4A,B). The RT-HDA assay was performed using T7 RNA (2.59 × 10^9^ to 2.59 × 10^2^ per reaction). The RT-HDA assay detected viral RNA at a dilution of 2.59 × 10^4^ per reaction with a reaction time of 90 min and a reaction temperature of 65 °C (Figure 4C,D). Furthermore, the RT-LAMP assay was performed using T7 RNA (2.33 × 10^10^ to 2.33 per reaction). The RT-LAMP assay detected viral RNA at a dilution of 2.33 × 10^2^ per reaction with a reaction time of 60 min and a reaction temperature of 65 °C (Figure 4E–G) using LFA. The negative control produced no signal at the test line. These results verify that the LAMP assay provided a relatively sensitive efficiency among these isothermal amplification assays. Thus, based on the efficiency of the isothermal amplification process, the LAMP with either LFA or colorimetric assay for SARS-CoV-2 naked-eye detection was identified as the optimal assay.

### 3.4. Optimization of the RT-LAMP Assay for SARS-CoV-2 Naked-Eye Detection

To optimize the LAMP-based assays for SARS-CoV-2 naked-eye detection, various LAMP primers were designed to target SARS-CoV-2 with S and N genes. We modified the FIP and BIP to include a “TTTT” linker between the F1c and F2 regions as well as between the B1c and B2 regions to further improve the reaction. The LAMP primers were assessed using synthesized viral RNA and WarmStart LAMP 2X Master Mix (NEB) and analyzed through the RT-LAMP, RT-LAMP with LFA, and colorimetric-LAMP assays. In the LAMP-LFA, positive samples clearly produced two colored bands at T (target) and C (control) that were readily distinguishable from the negative controls (main band at C). We observed that the LAMP primers targeting the S gene detected viral RNA (5.85 × 10^9^ to 5.85 × 10^6^ per reaction) through the RT-LAMP with LFA (Appendix A), colorimetric-LAMP (Appendix A), and on a 2% agarose gel (Appendix A) performed for 60 min at 65 °C. Furthermore, we observed that the LAMP primers targeting the N gene could detect the viral RNA through the RT-LAMP with LFA (Appendix A), colorimetric-LAMP (Appendix A), and on a 2% agarose gel (Appendix A) performed for 60 min at 65 °C.

To improve the sensitivity of the RT-LAMP reaction for SARS-CoV-2 detection, we evaluated the optimization of the primers and reaction time. The optimal reaction time was assessed using the RT-LAMP, colorimetric-LAMP, and RT-LAMP with LFA, viral RNA, and the LAMP primer set to target the S gene. Both the RT-LAMP and RT-LAMP with LFA detected the viral RNA (5.85 × 10^9^ to 5.85 × 10^7^ per reaction) at a reaction time of 30 or 60 min at 65 °C, respectively (Figure 5B–E). Moreover, the colorimetric-LAMP assay detected the viral RNA (5.85 × 10^9^ to 5.85 × 10^7^ per reaction) at a reaction time of 30 min or 60 min at 65 °C (Figure 5C,F). Furthermore, the resulting electrophoresis data showed viral RNA detection under the same conditions (Figure 5D,G). To further optimize the conditions for the S gene primers, loop primers (LF and LB) were subsequently evaluated using synthesized viral RNA through the RT-LAMP, RT-LAMP with LFA, and colorimetric-LAMP assays with synthesized viral RNA. The RT-LAMP with LFA, colorimetric-LAMP, and electrophoresis assays using only loop primers targeting the S gene did not amplify the viral RNA at a reaction time of 60 min and reaction temperature of 65 °C (Figure 5H–J). These results indicate that the LAMP primer set with loop primers is more sensitive than the loop primers alone.

### 3.5. Validation of COVID-19 Molecular Diagnostic System on Human Specimens

To evaluate the clinical utility of the COVID-19 molecular diagnostics system for SARS-CoV-2 detection, we tested 23 nasopharyngeal specimens including 8 patients with SARS-CoV-2 infection, 8 healthy participants, and 7 patients with other types of HCoV infection. The clinical specimens were processed for virus sample preparation using the DMP-PVDF filter platform. Then, the SARS-CoV-2 infection specimens were confirmed via the RT-LAMP with LFA and colorimetric RT-LAMP assays using the primer set for the S gene. We observed that the RT-LAMP with LFA could detect the 8 patients with SARS-CoV-2-positive specimens, as evidenced by a positive band on the flow strip (Figure 6A, upper). However, the colorimetric RT-LAMP assay could detect only 2 of the positive specimens (Figure 6A, lower). These results show that the RT-LAMP with LFA is more sensitive than the colorimetric RT-LAMP in the assessment of clinical specimens. The 8 negative controls from the healthy participants did not test positive via either the RT-LAMP with LFA or colorimetric RT-LAMP assays (Figure 6B). Furthermore, to validate the specificity of the RT-LAMP reaction for SARS-CoV-2 detection, we examined 7 nasopharyngeal specimens including 5 patients with an HCoV-OC43 infection and 2 patients with an HCoV-229E infection. The RT-LAMP with LFA, colorimetric-LAMP, and electrophoresis assays using LAMP primers targeting the S gene were not amplified for any of the viral RNA from the other types of HCoV specimens (Figure 6C). These results indicate that the RT-LAMP with LFA can be used for the detection of SARS-CoV-2-positive samples with the naked-eye at local POC centers. Therefore, the rapid COVID-19 molecular diagnostic system can be a useful tool for the diagnosis of COVID-19 at POCs via effective virus enrichment and extraction.

## 4. Conclusions

Previous high-quality studies have reported that isothermal amplification technologies can be used for SARS-CoV-2 detection from the RNA of clinical specimens, such as nasopharyngeal or throat swabs (Table 1). However, the diagnosis assays developed in these studies have limitations involving the kits and centrifugation steps [21,24,25,26] and require sophisticated instrumentation and special reagents, such as fluorescent dyes [22,24,25,26]. The cost of the proposed COVID-19 diagnostic system is much cheaper ($ 10.148/test) than that of the hospital test based on PCR ($ 136.18/test). Therefore, such assays involving isothermal amplification can be performed only in diagnostic facilities and cannot serve toward at-home or on-site testing, such as POC testing at emergency facilities and screening at airports.

In this study, we report a rapid COVID-19 molecular diagnostic system that can easily be operated using portable materials without any need for highly trained professionals or complex procedures. We demonstrate a diagnostic system that combines the RT-LAMP with a simple pathogen enrichment method involving a DMP-PVDF filter for rapid and accurate detection of SARS-CoV-2. Using the system, we validate the clinical utility in 23 clinical nasopharyngeal specimens and establish increased sensitivity and specificity. Moreover, the use of this proposed system for pathogen enrichment, amplification, and detection of viral RNA does not require expensive reagents. Thus, our results suggest that the combination RT-LAMP and DMP-PVDF filter system can be used for diagnostics purposes under limited-resource settings, at POCs, or even at home. The ultimate purpose of this proposed system is for early diagnosis of COVID-19 made directly to individuals at POCs or home, without any sophisticated instrumentation. While our findings confirm the utility of RNA extraction from enriched viruses through our new filter process, our new method requires further verification in clinical specimens to improve its versatility and ensure the utility of this system in clinical applications. We validated the COVID-19 molecular diagnostic system by using a limited number of human specimens in a clinical study. Therefore, further research is required to confirm the clinical utility of the COVID-19 molecular diagnostic system by using a large cohort of human specimens. In conclusion, the system described in this study provides a simple, sensitive, facile, efficient, and inexpensive strategy that can easily be applied for the diagnosis of emerging infectious diseases, such as COVID-19.

## Figures and Tables

**Figure 1 biosensors-11-00373-f001:**
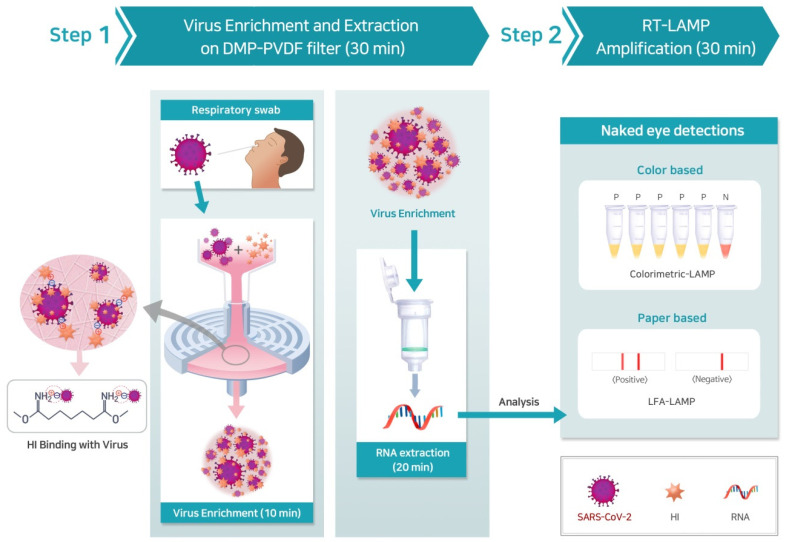
Schematic representation of the principle of a rapid COVID-19 diagnostic system through combining RT-LAMP and a DMP-PVDF filter platform. (Step 1) Schematic diagram of pathogen enrichment by DMP on PVDF filter. The capture of the pathogen by DMP via electrostatic interactions (**left**). The amine group of DMP enables the pathogen to be enriched on the PVDF membrane surface after 10 min of incubation (**right**). NA is extracted from the enriched pathogen. (Step 2) The extracted NA is visually confirmed by the naked-eye to pathogen detection via colorimetric RT-LAMP (top path) and RT-LAMP with LFA (bottom path). RT, reverse transcription; LAMP, loop-mediated isothermal amplification; DMP, dimethyl pimelimidate; PVDF, polyvinylidene fluoride; NA, Nucleic acid; LFA, lateral flow assay.

**Figure 2 biosensors-11-00373-f002:**
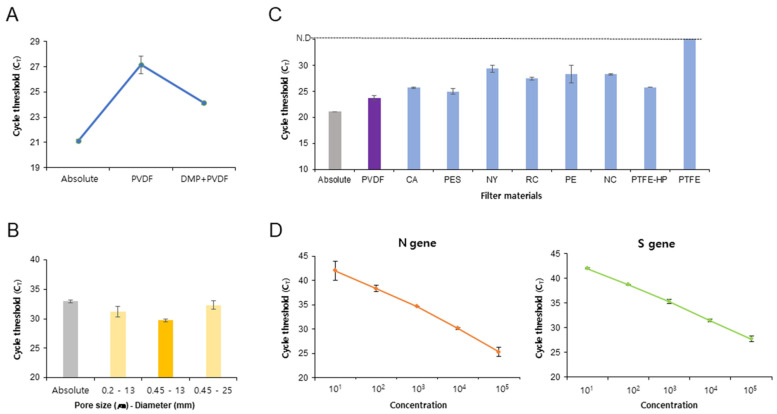
Application of DMP-PVDF filter platform to pathogen enrichment and extraction. (**A**) The amplification efficiency for a sample of *E. coli* (10^5^ CFU/mL) following bacterial enrichment and extraction using the DMP-PVDF platform is dependent on DMP. The positive control is DNA extracted from 10^5^ CFU of *E. coli* using a Qiagen kit. The data are presented as mean ± SD, based on at least three independent experiments. (**B**) The *E. coli* capture rate of the DMP-PVDF platform depends on the filter membrane’s pore size and diameter. The positive control (gray) is *E. coli* DNA extracted from 10^5^ CFU of *E. coli* using a Qiagen kit. The nearest and safest value to control (0.45 μm–13 mm) is marked in deep yellow. (**C**) The *E. coli* capture rate of the new platform depends on the type of filter membrane. The PVDF membrane (purple) shows the nearest and safest value to control. (**D**) Capacity of the DMP-PVDF filter platform to process test SARS-CoV-2 culture fluid ranging from 1 × 10^1^ to 1 × 10^5^ PFU/100 µL using a Qiagen kit. SARS-CoV-2 viral loads were quantified using RT-qPCR assay targeting the N and S genes. The data are presented as mean ± SD, based on at least three independent experiments. DMP, dimethyl pimelimidate; PVDF, polyvinylidene fluoride; N, nucleocapsid; S, spike; SD, standard deviation.

**Figure 3 biosensors-11-00373-f003:**
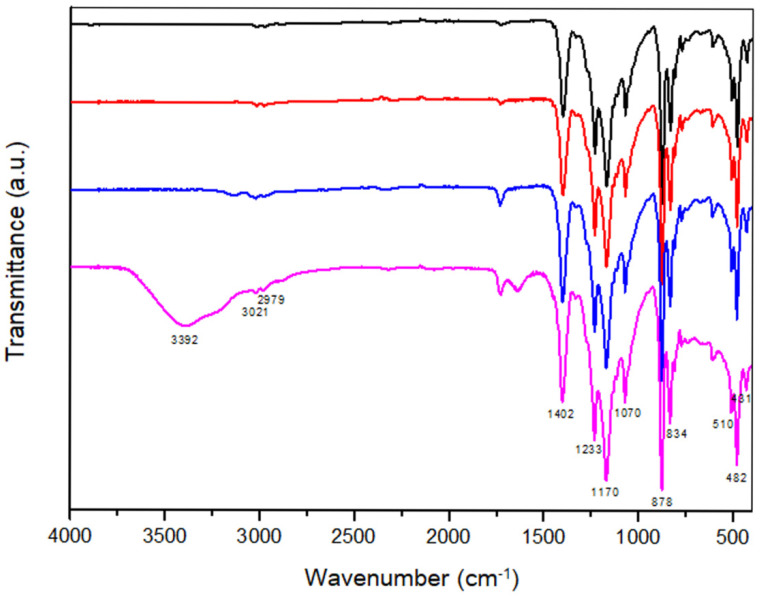
Fourier-transform infrared spectra analysis. The FTIR spectra of the plain PVDF membrane (black line), NAs (red line), DMP (blue line), and combination of the NAs and DMP (pink line) on the PVDF membrane are indicated. PVDF, polyvinylidene fluoride; NA, nucleic acid; DMP, dimethyl pimelimidate.

**Figure 4 biosensors-11-00373-f004:**
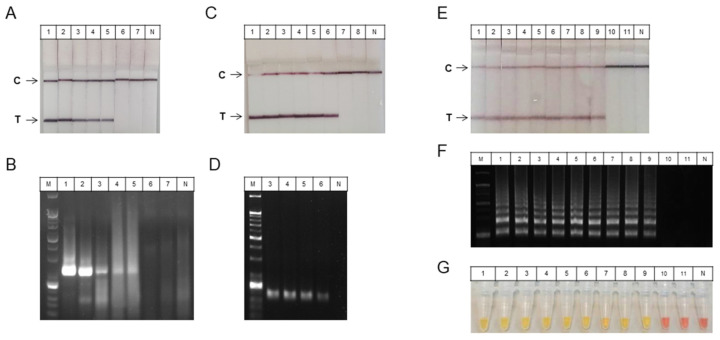
Sensitivities of RT-RPA, RT-HDA, and RT-LAMP assays for SARS-CoV-2 detection. A serial dilution of T7 RNA from the SARS-CoV-2 N gene was used to test the sensitivity of the assay. (**A**,**B**) The sensitivity of the RT-RPA assay. The RT-RPA reaction products could be detected by the T7 RNA (8.12 × 10^7^ to 8.12 × 10^3^ per reaction) on a (**A**) LFA and (**B**) 2% agarose gel. The sensitivity was 8.12 × 10^3^ of T7 RNA. (**C**,**D**) The sensitivity of the RT-HDA assay. The RT-HDA reaction products could be detected by the T7 RNA (2.59 × 10^9^ to 2.59 × 10^4^ per reaction) on a (**C**) LFA and (**D**) 2% agarose gel. Sensitivity was 2.59 × 10^4^ of T7 RNA. (**E**–**G**) The sensitivity of the RT-LAMP assay. The RT-LAMP reaction products visualized the T7 RNA (2.33 × 10^10^ to 2.33 × 10^2^ per reaction) on a (E) LFA, (**F**) 2% agarose gel, and (**G**) colorimetric RT-LAMP. Sensitivity was 2.33 × 10^2^ of T7 RNA. RT, reverse transcription; RPA, recombinase-based polymerase amplification; HDA, helicase-dependent amplification; LAMP, loop-mediated isothermal amplification; N, nucleocapsid; LFA, lateral flow assay; M, size marker; N; negative control; T, target; C, control.

**Figure 5 biosensors-11-00373-f005:**
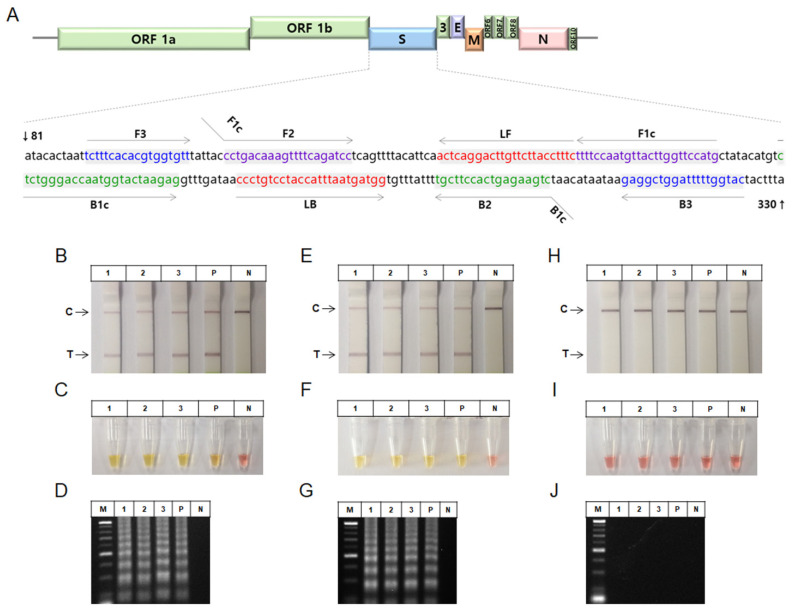
Sensitivity of RT-LAMP assay for SARS-CoV-2 detection. (**A**) SARS-CoV-2 S gene sequences and RT-LAMP primer designing. Available sequences were aligned to design primer sets that target the S gene. The locations that LAMP primers (F3, B3, FIP, BIP, LF, and LB) recognize on the S gene were highlighted. (**B**–**J**) RT-LAMP assay was performed using synthetic RNA (5.85 × 10^9^ to 5.85 × 10^7^ per reaction) for the S gene of SARS-CoV-2. (**B**–**G**) Assessment of the sensitivity of the RT-LAMP assay using a LAMP primer set comprising F3, B3, FIP, BIP, LF, and LB primers that recognize the S gene of SARS-CoV-2. RT-LAMP assay was performed for (**B**–**D**) 30 and (**E**–**G**) 60 min at 65 °C. (**B**,**E**) RT-LAMP with LFA, (**C**,**F**) colorimetric RT-LAMP, and (**D**,**G**) agarose gel electrophoresis of the reaction products. (**H**–**J**) Assessment of the sensitivity of the RT-LAMP assay by using only LF and LB primers. The RT-LAMP assays were performed at 65 °C. (**H**) RT-LAMP with LFA, (**I**) colorimetric RT-LAMP, and (**J**) agarose gel electrophoresis of the reaction products. RT, reverse transcription; LAMP, loop-mediated isothermal amplification; S, spike; LFA, lateral flow assay; M, size marker; P, positive control; N; negative control; T, target; C, control.

**Figure 6 biosensors-11-00373-f006:**
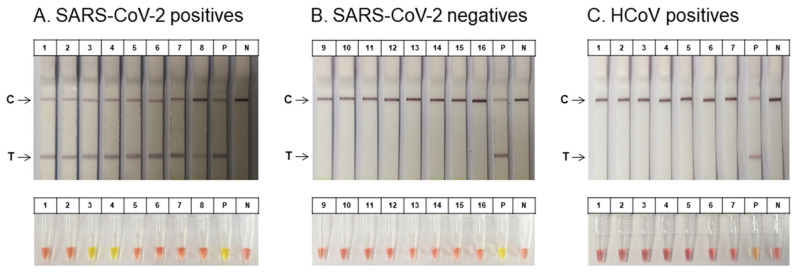
Clinical application of RT-LAMP and the DMP-PVDF filter system on human specimens. Clinical application of the RT-LAMP and DMP-PVDF filter system were analyzed for 23 human nasopharyngeal specimens with or without SARS-CoV-2 infection for the S gene of SARS-CoV-2 detection. (**A**) Eight patient specimens with SARS-CoV-2 infection were analyzed using the RT-LAMP and DMP-PVDF filter system by RT-LAMP with LFA (upper) and colorimetric RT-LAMP (lower). (**B**) Eight healthy participant specimens without SARS-CoV-2 infection were analyzed using the RT-LAMP with DMP-PVDF filter system by RT-LAMP with LFA (upper) and colorimetric RT-LAMP (lower). (**C**) Specificity of RT-LAMP assay for SARS-CoV-2 detection. (**C**) RT-LAMP assay was performed using 7 evaluated clinical nasopharyngeal specimens from 5 HCoV-OC43 (*lanes* 1–5) and 2 HCoV-229E (*lanes* 6 and 7) patients for the S gene of SARS-CoV-2. The RT-LAMP assays were performed at 65 °C. RT-LAMP with LFA (upper) and colorimetric RT-LAMP (lower). RT, reverse transcription; LAMP, loop-mediated isothermal amplification; S, spike; LFA, lateral flow assay; P, positive control; N; negative control; T, target; C, control.

**Table 1 biosensors-11-00373-t001:** Comparison among different isothermal amplification technologies for SARS-CoV-2 detection of clinical specimens.

Name	Category	Platform	Types of Clinical Specimens	Gene	Available Readouts	Time for Amplification and Detection	LOD ^a^	Ref
Rapid COVID-19 diagnostic system	LAMP-based methods	Rapid platform	23 nasopharyngeal swabs	S or N	LFA, Colorimetric RT-LAMP	40–60 min	5	This study
DETECTR assay		CRISPR-Cas12	83 respiratory swabs	N and E	LFA	30–40 min	10	[21]
opvCRISPR assay		CRISPR-Cas12a	50 nasopharyngeal swabs	S	The reaction tube under illuminator	45 min	5	[22]
SHERLOCK assay	RPA-based methods	CRISPR-Cas13	534 nasopharyngeal and throat swabs	S and N	Fluorescence, LFA	35–70 min	42	[24]
AIOD-CRISPR assay		CRISPR-Cas12a	28 swabs	N	The reaction tube under transilluminator	20 min	5	[25]
CRISPR-FDS assay		CRISPR-Cas12a	29 nasal swabs	N and ORF1ab	RT-qPCR	50 min	5	[26]

LAMP, loop-mediated isothermal amplification; RPA, recombinase-based polymerase amplification; S, spike; N, nucleocapsid; ORF, open reading frame; RT, reverse transcription; qPCR, quantitative polymerase chain reaction; LFA, lateral flow assay; LOD, limit of detection; **^a^** Standardized LOD (minimum detectable viral RNA copies/reaction).

## Data Availability

Not applicable.

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
