# Peer review of "Rapid COVID-19 Molecular Diagnostic System Using Virus Enrichment Platform"

_biosensors, 2021, doi:10.3390/bios11100373_

Round 1
Reviewer 1 Report
The authors report on a rapid molecular diagnostic system that combines a simple virus enrichment and extraction method involving DMP, PVDF filter, and LAMP-LFA for the rapid (only 60 minutes), point-of-care (!) and sensitive detection of SARS-CoV-2. The test system proposes self-powered sample preparation assay with naked-eye detection without the use of any sophisticated instrumentation. Acknowledging the importance of rapid and reliable diagnostics of SARS-CoV-2 and the drawbacks of conventional PCR-based approach, the study raises big interest for every day praxis in the time of COVID-19 pandemic.
As a physician working with COVID-19 patients I found this study and approach very important and interesting. Indeed, the rapid and reliable point-of care diagnostics has a huge importance in a clinical daily routine. The paper is well written and organized that is why I have just a few comments:
- Only 8 patients with SARS-CoV-2 were tested by the proposed approach. These should be commented as a limitation of the study in the discussion.
- What is the plan considering application of this approach in a real life-scenario in the future? I would give a reader some perspectives in the discussion. E.g. comparison of results with a conventional PCR based-approach in a Hospital.
- What are the costs of these approach? Can they be compared with costs of other test systems (e.g. PCR)?
- The approach is well described and presented giving an opportunity not only biologists but also medical stuff to understand the main principles.
- The paper is extensively referenced, which is informative.
- The figures and tables are informative and well structured.
Author Response
We sincerely appreciate the Reviewers for their thoughtful review of our manuscript. We have taken their comments into careful consideration in preparing our revision, significantly improving the quality of the manuscript in the process. Below, we present point-by-point responses to the reviewers’ comments.
Reviewer #1: The authors report on a rapid molecular diagnostic system that combines a simple virus enrichment and extraction method involving DMP, PVDF filter, and LAMP-LFA for the rapid (only 60 minutes), point-of-care (!) and sensitive detection of SARS-CoV-2. The test system proposes self-powered sample preparation assay with naked-eye detection without the use of any sophisticated instrumentation. Acknowledging the importance of rapid and reliable diagnostics of SARS-CoV-2 and the drawbacks of conventional PCR-based approach, the study raises big interest for everyday praxis in the time of COVID-19 pandemic.
As a physician working with COVID-19 patients I found this study and approach very important and interesting. Indeed, the rapid and reliable point-of care diagnostics has a huge importance in a clinical daily routine. The paper is well written and organized that is why I have just a few comments:
We sincerely appreciate the reviewer for very carefully reading the manuscript, the constructive comments, suggestions, which have helped us significantly improve the quality of the manuscript.
- Only 8 patients with SARS-CoV-2 were tested by the proposed approach. These should be commented as a limitation of the study in the discussion.
Response: We thank the reviewer for this comment. We agree with your comment. First, we blind tested the accuracy of our COVID-19 diagnostic system by using 23 clinical nasopharyngeal specimens suspected of being infected with SARS-CoV-2. Next, we confirmed the COVID-19 diagnosis results using our system and compared them with the clinical patient diagnoses in a hospital. Then, we confirmed that 8 out of 23 clinical nasopharyngeal specimens were SARS-CoV-2 positive specimens.
According to the reviewer’s comment, we have mentioned this limitation of the study in terms of patient numbers in the revised manuscript as follows:
On page 12-13, “We validated the COVID-19 molecular diagnostic system by using a limited number of human specimens in a clinical study. Therefore, further research is required to confirm the clinical utility of the COVID-19 molecular diagnostic system by using a large cohort of human specimens.”
- What is the plan considering application of this approach in a real life-scenario in the future? I would give a reader some perspectives in the discussion. E.g. comparison of results with a conventional PCR based-approach in a Hospital.
Response: We thank the reviewer for this comment. The ultimate purpose of this study using combining the RT-LAMP and DMP-PVDF filter system is for rapid diagnosis under the limited-resource settings such as POCs, home. If someone has a fever, cough, or other symptoms, we think it is better to perform the COVID-19 diagnostic test at POCs or at home rather than for them to go to the hospital right away.
In terms of efficiency, going to the hospital after pre-testing as positive for COVID-19 would improve efficiency. Hence, the ultimate purpose is early diagnostics directly to individuals around the world in POC testing, requiring a self-sample preparation without any sophisticated instrumentation. According to the reviewer’s comment, we have mentioned it in the revised manuscript as follows:
On page 12, “The ultimate purpose of this propose system is for early diagnosis of COVID-19 directly to individuals at POCs or home, without any sophisticated instrumentation.”
- What are the costs of these approach? Can they be compared with costs of other test systems (e.g. PCR)?
Response: We thank the reviewer for this comment. We have calculated the costs of reagents used by the proposed COVID-19 diagnostic system, and these are presented in the table below.
|
No. |
Reagent |
Cat. no |
Cost ($) |
|
1 |
Syringe PVDF filter |
FJ13BNPPV004AD01 |
1.08 / test |
|
2 |
Syringe (2mL) |
|
0.048 / test |
|
3 |
DMP |
D-8388 (Sigma) |
1.17 / test |
|
4 |
QIAGEN viral kit |
QIAGEN 52906 |
5.98 / test |
|
5 |
LAMP reagent |
BioLabs, M1800 |
1.87 / test |
|
Total cost |
$ 10.148 / test |
||
The price per use of our COVID-19 diagnostic system is about $ 10.148 per human sample. However, the price of testing at hospitals using PCR system in Republic of Korea is $ 136.18. Moreover, the price of testing at hospitals in the United States is over $ 1447. Thus, the proposed COVID-19 diagnostic system is much cheaper and more efficient than hospital tests. According to the reviewer’s comment, we have mentioned it in the revised manuscript as follows:
On page 12, “The cost of the proposed COVID-19 diagnostic system is much cheaper ($ 10.148/test) than that of the hospital test based on PCR ($ 136.18/test).”
- The approach is well described and presented giving an opportunity not only biologists but also medical stuff to understand the main principles.
Response: We thank the reviewer for this comment.
- The paper is extensively referenced, which is informative.
Response: We thank the reviewer for this comment.
- The figures and tables are informative and well structured.
Response: We thank the reviewer for this comment.
Reviewer 2 Report
Dear editor and authors!
The article on title "Rapid COVID-19 molecular diagnostic system using virus enrichment platform" is a new and interesting. The authors present in the article report a rapid COVID-19 molecular diagnostic system that combines a self-powered sample preparation assay and naked-eye detection method for the rapid and sensitive detection of SARS-CoV-2. I recommend a minor revision.
The manuscript is well written, thought out and laid out. I find the article need a small improves, but after that’s I think it will be to the liking of readers Biosensors.
I consider following improvements necessary before publication:
General comments (paragraphs):
25: “naked-eye detection of SARS-CoV-2” clarify, please
32: “The validated the accuracy of the system by using 23 clinical nasopharyngeal specimens.” In my opinion the accuracy of the new systems should checked by using at least 100 clinical samples or more.
90: The technique should be mentioned in the abstract ((RT)-LAMP assay) because the presented system is very enigmatic up to the 90th paragraph.
246: In addition to visual reading, do you consider spectrophotometric reading functions for a specific wavelength?
Regards, R1
Author Response
We sincerely appreciate the Reviewers for their thoughtful review of our manuscript. We have taken their comments into careful consideration in preparing our revision, significantly improving the quality of the manuscript in the process. Below, we present point-by-point responses to the reviewers’ comments.
Reviewer #2: The article on title "Rapid COVID-19 molecular diagnostic system using virus enrichment platform" is a new and interesting. The authors present in the article report a rapid COVID-19 molecular diagnostic system that combines a self-powered sample preparation assay and naked-eye detection method for the rapid and sensitive detection of SARS-CoV-2. I recommend a minor revision.
The manuscript is well written, thought out and laid out. I find the article need a small improves, but after that’s I think it will be to the liking of readers Biosensors.
I consider following improvements necessary before publication:
We sincerely appreciate the reviewer for very carefully reading the manuscript, the constructive comments, suggestions, which have helped us significantly improve the quality of the manuscript.
General comments (paragraphs):
- 25: “naked-eye detection of SARS-CoV-2” clarify, please
Response: We thank the reviewer for this comment. The “naked-eye detection of SARS-CoV-2” in this study means RT-LAMP with LFA and colorimetric RT-LAMP.
Especially in the resources-limit region, conveniently operated detection methods such as “naked-eye” detection are urgently required that no instrument is needed. Amongst these, colorimetric assays have attracted widespread attention due to their simplicity, cost-effectiveness, no need for complex instruments, and especially easy to read out with the naked-eye, which may provide new opportunities for SARS-CoV-2 detection. Since RT-LAMP is an isothermal process (usually at 60 °C to 65 °C), the assay requires only a simple instrument (e.g., water bath or heating block). In addition, it demands short reaction times (within an hour) and, in the presence of a colorimetric pH indicator (e.g. phenol red or crystal violet) or a complexometric dye (e.g. SYBR green or calcein), the results can be visualized by the naked eye. Therefore, compared to qRT-PCR, RT-LAMP is a more appealing choice for the high throughput, low-cost detection of SARS-CoV-2. The RT-LAMP product formation was visualized by the naked eye.
- 32: “The validated the accuracy of the system by using 23 clinical nasopharyngeal specimens.” In my opinion the accuracy of the new systems should checked by using at least 100 clinical samples or more.
Response: We thank the reviewer for this comment. We agree with your comment, but it was very difficult to obtain clinical specimens. Therefore, we could only perform our COVID-19 diagnostic system using a small number of clinical samples. According to the reviewer’s comment, we have mentioned this limitation of the study in terms of the number of human specimens in the revised manuscript as follows:
On page 12-13, “We validated the COVID-19 molecular diagnostic system by using a limited number of human specimens in a clinical study. Therefore, further research is required to confirm the clinical utility of the COVID-19 molecular diagnostic system by using a large cohort of human specimens.”
- 90: The technique should be mentioned in the abstract ((RT)-LAMP assay) because the presented system is very enigmatic up to the 90th paragraph.
Response: We thank the reviewer for this comment. We agree with your comment. According to the reviewer’s comment, we have mentioned it in the revised manuscript as follows:
On page 1, “Here, we report a rapid COVID-19 molecular diagnostic system that combines a self-powered sample preparation assay and loop-mediated isothermal amplification (LAMP) based naked-eye detection method for the rapid and sensitive detection of SARS-CoV-2. The self-powered sample preparation assay with a hydrophilic polyvinylidene fluoride filter and dimethyl pimelimidate can be operated by hand, without the use of any sophisticated instrumentation, similar to the re-verse transcription (RT)-LAMP-based lateral flow assay, for the naked-eye detection of SARS-CoV-2.”
- 246: In addition to visual reading, do you consider spectrophotometric reading functions for a specific wavelength?
Response: We thank the reviewer for this comment. The purpose of our study was to diagnose COVID-19 without a specific device. Therefore, we did not consider a detection method that requires additional equipment, including a spectrophotometer reading instrument for a specific wavelength.

Reviewer 3 Report
In the present manuscript, the authors try to generate a new modification in the protocols for molecular diagnostic of COVID-19. Pleasantly the experiments are performed correctly, and the findings are of good quality. Moreover, the idea of generating a platform to enrich the amount of virus in the sample brings a better implementation of it for the tests, thereby improving its sensitivity and quality, for which I recommend that the manuscript be accepted only with the following modifications.
The principal idea of the authors is to improve the virus enrichment previous to extraction; interestingly, the authors performed a relevant control with bacteria; after used the purified system of QIAGEN and compared the process of pathogen enrichment. The authors must show this experiment with RNA viral and quantified this RNA in condition enrichment or not.
Figure 3 is very confusing; in the plot, the colors of the lines cannot be differentiated; I recommend modifying the format and a table where they describe what values are obtained in their spectrum.
Author Response
We sincerely appreciate the Reviewers for their thoughtful review of our manuscript. We have taken their comments into careful consideration in preparing our revision, significantly improving the quality of the manuscript in the process. Below, we present point-by-point responses to the reviewers’ comments.
Reviewer #3: In the present manuscript, the authors try to generate a new modification in the protocols for molecular diagnostic of COVID-19. Pleasantly the experiments are performed correctly, and the findings are of good quality. Moreover, the idea of generating a platform to enrich the amount of virus in the sample brings a better implementation of it for the tests, thereby improving its sensitivity and quality, for which I recommend that the manuscript be accepted only with the following modifications.
We sincerely appreciate the reviewer for very carefully reading the manuscript, the constructive comments, suggestions, which have helped us significantly improve the quality of the manuscript.
- The principal idea of the authors is to improve the virus enrichment previous to extraction; interestingly, the authors performed a relevant control with bacteria; after used the purified system of QIAGEN and compared the process of pathogen enrichment. The authors must show this experiment with RNA viral and quantified this RNA in condition enrichment or not.
Response: We thank the reviewer for this comment. To evaluate the usability of the DMP-PVDF filter for the pathogen enrichment and extraction, we evaluated its capacity to detect E.coli for DNA and the Brucella ovis for RNA ranging from 1 × 102 to 1 × 105 CFU/mL (Figure S1B and S1C). The RT-qPCR CT value of the DNA and RNA extracted using the DMP-PVDF filter was earlier than those obtained using the spin column kit, which extracted DNA and RNA without pathogen enrichment (Figure S1B and S1C). Moreover, the amplification efficiency of the DMP-PVDF filter platform was similar to the absolute value of the pathogen (Qiagen kit) (Figure S1B and S1C). These results suggest that the capacity of the DMP-PVDF platform with pathogen enrichment is more efficient than those of conventional commercial kits that do not involve pathogen enrichment. Next, we investigated the capacity of the DMP-PVDF platform with pathogen enrichment by using SARS-CoV-2 culture fluid ranging from 1 × 101 to 1 × 105 PFU/100µL. SARS-CoV-2 viral loads were quantified using RT-qPCR assay for targeting the N and S genes (Figure 2D). The detection limit of the platform is 10 PFU/100µL for both N and S genes. These results suggest that the DMP-PVDF platform can also be used with SARS-CoV-2 enrichment. Although we did not perform the comparison between the DMP-PVDF platform and the QIAGEN kit using SARS-CoV-2 culture fluid, we conducted the SARS-CoV-2 viral loads for clinical specimens were quantified by real-time RT-PCR assay. We used the correlation curves of the N gene (left) and the S gene (right) generated by linear regression plots of the cycle threshold values. The limit of detection for SARS-CoV-2 was 5 copies per reaction.
N gene S gene
- Figure 3 is very confusing; in the plot, the colors of the lines cannot be differentiated; I recommend modifying the format and a table where they describe what values are obtained in their spectrum.
Response: We thank the reviewer for this comment. According to the reviewer’s comment, we have modified Figure 3 and its legend in the revised manuscript.
Figure 3. Fourier-transform infrared spectra analysis. The FTIR spectra of the plain PVDF membrane (black line), NAs (red line), DMP (blue line), and combination of the NAs and DMP (pink line) on the PVDF membrane are indicated.
